# A Randomized Computer-Assisted Rehabilitation Trial of Attention in Pediatric Multiple Sclerosis: A Post Hoc Analysis

**DOI:** 10.3390/brainsci11050637

**Published:** 2021-05-14

**Authors:** Marta Simone, Rosa Gemma Viterbo, Lucia Margari, Pietro Iaffaldano

**Affiliations:** 1Child Neuropsychiatry Unit, Department of Biomedical Sciences and Oncology, University of Bari “Aldo Moro”, 70121 Bari, Italy; lucia.margari@uniba.it; 2MS Centre, Department of Basic Medical Sciences, Neurosciences and Sense Organs, University of Bari “Aldo Moro”, Piazza G. Cesare, 11, 70121 Bari, Italy; rossellaviterbo4@gmail.com (R.G.V.); pietro.iaffaldano@uniba.it (P.I.)

**Keywords:** pediatric multiple sclerosis, Symbol Digit Modalities Test, cognitive impairment, attention, rehabilitation

## Abstract

Cognitive impairment (CI) is a remarkable feature in pediatric-onset multiple sclerosis (POMS). The Symbol Digit Modalities Test (SDMT) is increasingly used to explore CI in MS. Recently, a four-point worsening on the SDMT score has been demonstrated to correlate with a clinically meaningful cognitive worsening in adult MS. We conducted a post hoc analysis of a randomized computer-assisted rehabilitation trial for attention impairment in POMS to test the clinical meaningfulness of the changes in SDMT scores at the end of the trial (delta SDMT). A four-point SDMT cut-off was applied. POMS patients exposed to specific computer training (ST) and non-specific training (nST) were compared. Data of 16 POMS (9 females, age 15.75 ± 1.74 years) patients were analyzed. At the end of the trial, 25% of patients reported no clinically significant changes (−3 to 3), 12.5% a clinically significant worsening (≤−4) and 62.5% a clinically significant improvement (≥4) in the delta SDMT. The proportion of patients reporting a clinically meaningful improvement was significantly (*p* = 0.008) higher (100%) in patients exposed to ST in comparison to those (25%) exposed to nST. The use of the four-point SDMT cut-off may be useful to assess the clinical meaningfulness of results from cognitive rehabilitation trials.

## 1. Introduction

There is a growing need to find new and more robust disability outcome measures to be used in multiple sclerosis (MS) randomized clinical trials (RCTs) and in clinical practice. The most common outcomes currently used, annualized relapse rate and sustained Expanded Disability Status Scale (EDSS) progression, miss an important dimension of MS-related disability, namely, the decline in cognitive function. Indeed, cognitive dysfunction is one of the most remarkable features of MS [1] and particularly in pediatric-onset MS (POMS) [2,3,4,5,6].

The percentage of patients with POMS with at least a mild cognitive deficit ranges from 30 to 50% [2,3,4,5,6]. Therefore, the Multiple Sclerosis Functional Composite (MSFC), including a cognitive test, the Paced Auditory Serial Addition Test (PASAT), has been suggested [7] as an alternative and more complete outcome measure in MS RCTs, but it has some limitations due to the difficulty in interpreting the clinical meaning of z-score changes and because it seems to not be fully accepted by MS patients [8,9].

Currently, the Symbol Digit Modalities Test (SDMT), a simple, brief measure of information processing speed (IPS), is considered the gold standard in screening for cognitive involvement in MS [10,11,12,13]. Furthermore, this test explores the cognitive domain collectively known as “attention”. In particular, it addresses the so-called “sustained attention” and “concentration”. This is possible throughout several cognitive processes which also include IPS, working memory and cognitive flexibility [10,11,12,13].

Due to its short duration and ease of administration, high sensitivity in detecting subtle changes in cognitive functioning in MS, good test–retest reliability and very low practice effects [10,11,12,13], routine administration of the SDMT has been adopted by many clinicians. Moreover, it has been proposed and is increasingly used to explore cognitive functions in MS clinical trials [10,11,12,13]. 

Recently, a detailed analysis of the psychometric qualities, sensitivity to change and clinical meaningfulness of the SDMT in comparison to the PASAT was performed by the Multiple Sclerosis Outcome Assessments Consortium (MSOAC) [14]. The results of this analysis proved the SDMT to be superior to the PASAT, suggesting the SDMT should be considered the measure of choice for MS trials in assessing IPS. In particular, they found that a four-point worsening on the SDMT score significantly correlated with clinically meaningful cognitive decline, as evidenced by a five-point worsening on the Physical Component Summary (PCS) of the Health Status Questionnaire (SF-36) [14]. Moreover, previous studies confirmed that this degree of change in the SDMT is clinically meaningful, when correlated to relapses and employment status [13,14,15,16]. In a recent double-blind RCT [17], we assessed the efficacy of a home-based computerized program for retraining attention dysfunction in a cohort of POMS patients with attention impairment. We found that after a 3-month cognitive training, the specific computer training (ST) exposure was associated with a significantly more pronounced reduction in a global measure of cognitive functioning, the Cognitive Impaired Index (CII), in comparison to the non-specific training (nST) exposure. In particular, POMS patients treated with ST had a significantly higher improvement in their performances on the SDMT in comparison to those receiving n-ST, suggesting that a cognitive rehabilitation program that targets attention is a suitable tool for improving global cognitive functioning in POMS patients. 

In fact, in our trial, we observed an improvement also in cognitive domains not specifically trained by the program [18]. Patients with POMS improve their executive functioning, planning strategies, visuo-spatial memory and delayed recall performances. These findings indicate that ST induces both a near transfer effect in the domain of the planning strategies and a far transfer effect in the domain of visuo-spatial memory. 

Here, we present a post hoc analysis aimed to assess the robustness of treatment effects, applying the four-point SDMT cut-off, as proposed by MSOAC [14], on the results of our cognitive rehabilitation trial.

## 2. Materials and Methods

A detailed description of the study population, procedures and intervention has been previously reported elsewhere [17].

Briefly, 16 POMS patients diagnosed according to the most recent diagnostic criteria, aged <18 years, with an Expanded Disability Status Scale (EDSS) score ≤5.5 and failing (defined as scores <1.5 standard deviation (SD) of normative values) in at least 2/4 attention tests on a neuropsychological battery, were randomized to ST or nST [19], performed at home, in one-hour sessions, twice/week for three months. 

ST targets focused, sustained, selective, alternating and divided attention and consists of a group of hierarchically organized tasks that exercise different components of attention, proceeding from sustained to selective, alternating and, finally, divided attention exercises. The sequence of the exercises places increasing demands on complex attention control and working memory systems.

A neuropsychological test battery was administered, using alternative versions of the tests, at baseline (T0), and within one week following the end of the three-month training program (T1). 

The neuropsychological test battery comprised tests which cover different cognitive domains: Selective Reminding Test (SRT), Selective Reminding Test–Delayed (SRT-D), Spatial Recall Test (SPART), Spatial Recall Test–Delayed (SPART-D), SDMT, Trail Making Tests (TMT) A and B, Semantic Verbal Fluency Test (SVFT) and Tower of London Test (TOL). In particular, the SDMT was administered to assess concentration, attention and IPS. A global score, the CII, allowing the evaluation of changes in cognitive performances independently by the number of cognitive tests failed at the neuropsychological evaluation, was obtained using the mean and SD from the normative values for each test. A detailed description of the CII calculation procedures has been extensively reported previously [17,19,20,21,22,23].

The primary objective of this post hoc analysis was to test the effects of our cognitive rehabilitation trial by comparing the delta SDMT scores, applying a 4-point SDMT cut-off, in ST and nST groups.

Statistical analysis:

All analyses were post hoc and not pre-specified. 

Continuous variables were described as mean and SD, and categorical variables were described as frequency and percentage. Group comparison was performed using Student’s t test, the Mann–Whitney U test and Fisher’s exact test, when appropriate.

As previously reported elsewhere [17], by using the mean and SD from the normative values for each test, we obtained a global cognitive score, the Cognitive Impairment Index (CII). This score allows the evaluation of changes in cognitive performances independently by the number of cognitive tests failed at the neuropsychological evaluation [17,19,21,22].

For each patient, a grading system was applied to individual cognitive tests, based on the number of SDs below the control mean (i.e., grade 0 was given if the patient scored at or above the control mean, and 1 if he/she scored below the control mean, but at or above 1 SD below the control mean, and so on until all patient scores were accommodated) [17,19,21,22]. The individual CII was obtained by the sum of all the patient’s scores. Moreover, to calculate the delta CII, it was calculated as CII score at T1—CII score at T0. The higher the improvement in the global cognitive score as measured by the CII at the end of the training (T1), the lower the delta CII. 

To quantify the clinical impact of changes over time of the SDMT in the cohorts of patients exposed to ST and to nST, we applied the following categorization of the delta SDMT scores (delta SDMT = SDMT score at T1—SDMT score at T0) by using a 4-point SDMT cut-off:− Delta SDMT between −3 and 3 = no clinically significant SDMT change;− Delta SDMT ≤−4 = clinically significant SDMT worsening;− Delta SDMT ≥ 4 = clinically significant SDMT improvement.

Thereafter, we compared the proportion of patients reporting a clinically significant SDMT improvement at the end of the 3-month training program by using the chi-square test.

Finally, to evaluate the relationship between changes in the SDMT scores and in the CII, we compared the delta CII among the groups of patients stratified on the basis of the delta SDMT (clinically/no clinically significant SDMT changes).

Statistical analysis was performed by using SPSS software (SPSS, version 22.0; SPSS, Chicago, IL, USA). 

## 3. Results

The comparisons of baseline demographic and clinical characteristics and of the baseline NP of POMS subgroups who underwent ST and n-ST are reported in Table 1. 

At baseline, no differences were found between the two treatment arms regarding sex and age and in terms of NP performances.

After the 3-month cognitive training, patients exposed to ST showed a significant improvement in SDMT performances (*p* < 0.0001) in comparison to those treated with nST.

In more detail, a significant effect for time (Baseline (T0) vs. Post-Treatment (T1) comparison) was found for the mean (SD) SDMT values (ST group: 24.5 (4.6) vs. 46.3 (6.7); nST group: 20.5 (3.6) vs. 20.8 (4.1), *p* < 0.0001) [17].

By applying the four-point cut-off of the delta SDMT scores, 4 (25%) patients reported no clinically significant changes, 2 (12.5%) patients a clinically significant worsening and 10 (62.5%) patients a clinically significant improvement in the SDMT score at the end of the training program. 

The proportion of patients reporting a clinically significant improvement in the SDMT was significantly (*p* = 0.008) higher in patients exposed to ST (8/8;100%) in comparison to that in patients exposed to nST (2/8; 25%) (Table 2). 

Among the other patients exposed to nST, 4/8 (50%) reported stable SDMT scores (delta SDMT between −3 and +3, meaning no clinically significant change), and 2/8 (25%) had a significant deterioration. 

Moreover, the overall improvement in the delta CII was significantly higher in patients reporting a clinically significant improvement in the SDMT at the end of training in comparison to those who presented with no clinically significant change and those with a clinically significant worsening of the SDMT (*p* = 0.038) (Figure 1).

## 4. Discussion

Rehabilitation treatment with a computerized cognitive training specifically designed to exercise the attention domain resulted in a significant improvement in overall cognitive performances and, in particular, in the SDMT scores. With this post hoc analysis, we have also demonstrated that a specific attention training is associated with clinically meaningful changes in SDMT scores in the short term.

It is noteworthy that all patients exposed to ST exhibited a clinically meaningful improvement in the SDMT scores in comparison to only 2/8 patients exposed to nST at the end of the 3-month cognitive training.

To the best of our knowledge, this is the first report of the application of the four-point SDMT cut-off, proposed by MSOAC [14], to the results of an RCT in order to test if the degree of changes in the SDMT scores obtained after a specific cognitive training was clinically meaningful in a pediatric cohort. 

The research about functional measurers capable of exploring from different perspectives, including non-motor disability, the overall impact of MS on disability has a long-lasting history.

In 1996, the MSFC was proposed as a multiple-domain measure to detect and summarize walking impairment (via the timed 25-foot walk test), upper extremity dexterity (via the nine-hole peg test) and cognition (via the PAST) abilities in patients with MS [24].

Thereafter, most of the RCTs performed to evaluate the efficacy of disease-modifying therapies included the MSFC as an outcome measure, but due to the difficulty in interpreting the clinical meaning of z-score changes and because it seems that MS patients do not fully accept the tests (especially the PASAT), the MSFC has not been extensively used in clinical practice [8,9].

The use of the SDMT as an outcome measure in RCTs and observational studies has progressively gained more attention in recent years [10,11,12,13,14]. The SDMT performances have been found to be associated with different magnetic resonance imaging measures of MS disease progression [25,26,27,28,29,30].

Moreover, SDMT scores are predictive of different patient-related outcomes, such as employment and driving abilities [31,32,33], but also of future cognitive decline [34]. 

Given all these premises and based on its predictive validity, the very high level of sensitivity and specificity and the facility of administration, this cognitive test is often used in clinical practice to perform a basic cognitive screening helping to identify patients at high risk for cognitive impairment who need a more structured neuropsychological evaluation. 

Recently, the psychometric properties of SDMT and PASAT have been compared and the former proved to be superior to the PASAT in assessing the IPS in patients with MS. [14]

Furthermore, a four-point change in the SDMT has been proven to be a cut-off able to discriminate clinically meaningful changes from test score changes due to a practice effect or simply chance [14]. In this post hoc analysis of a randomized trial, we showed, for the first time in a pediatric cohort, that an increase of at least four points in the SDMT could also indicate a clinically meaningful improvement in cognitive functionating. This is based on the observation that all the patients with an increase in the SDMT score of at least four points also reported an improvement in their overall cognitive functioning as measured by the CII.

Different limitations of this study deserve discussion. First, due to the monocentric nature of the study and due to the absence of external funding resources, the recruitment was limited to 16 POMS patients. POMS is a rare disease; only 5% of MS patients have an onset before 18 years, and only some of them present cognitive deficits which represent the main inclusion criteria of our trial. The internal validity and the consistency of our results are assured by a randomized controlled design and robust statistical analysis.

Second, we did not include in the study protocol the collection of measures of clinically meaningful changes such as school performance or parent-reported behaviors. This was a pilot study designed for a fast evaluation of the efficacy of a cognitive training (in treated vs. untreated patients) in POMS patients with attention impairment. Anyhow, we were able to show that a cognitive training specifically addressed to stimulate different attention components resulted in an important transfer effect. 

## 5. Conclusions

In conclusion, in our RCT, the use of the four-point SDMT cut-off allows us to demonstrate the clinical meaningfulness and the robustness of the treatment effects obtained by a home-based computerized program for retraining attention dysfunction in POMS patients with attention impairment. 

Further studies on larger populations are needed to confirm the clinical validity of this cut-off and its applicability in the routine clinical practice setting. 

## Figures and Tables

**Figure 1 brainsci-11-00637-f001:**
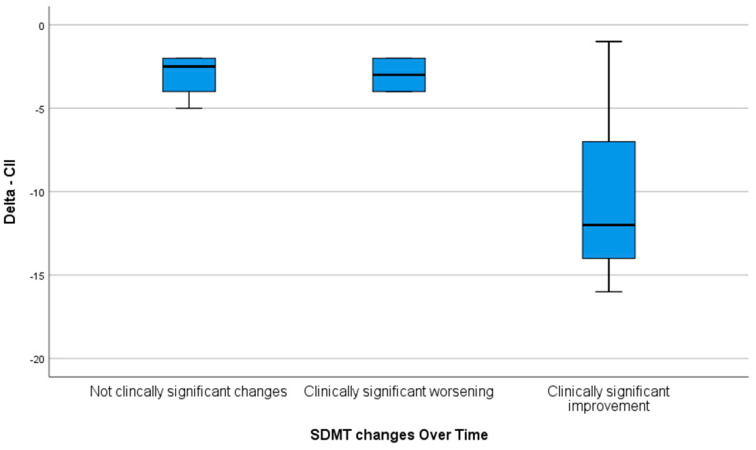
Delta CII at the end of the training stratified by the delta SDMT score.

**Table 1 brainsci-11-00637-t001:** Baseline demographic and clinical characteristics of POMS subgroups who underwent specific and non-specific training.

Variable	Specific Training (*n* = 8)	Non-Specific Training (*n* = 8)	*p*-Value (*t*, U or Fisher’s Exact Test)
Sex (F/M)	5/3	4/4	1.0
Age, years	15.8 (2.0)	15.7 (1.5)	1.0
Disease duration, years	3.5 (3.5)	3.3 (2.6)	0.96
Handedness, n. right-handed (%)	7 (87.5)	8 (100)	0.97
Disease-modifying therapy, n			
Nothing	2	2	0.67
Interferon beta	6	4	
Glatiramer acetate	0	1	
Natalizumab	0	1	
Annualized relapse rate	0.4 (0.5)	0.3 (0.5)	0.72
EDSS, median (min–max)	2.0 (1.0–3.5)	3.0 (1.0–3.5)	0.28
**Baseline Neuropsychological Performances**			
SRT-LTS	29.9 (12.6)	24.6 (6.5)	0.2
SRT-CLTR	22.1 (11.0)	20.4 (7.5)	0.6
SPART	19.3 (4.4)	22.8 (2.0)	0.1
SDMT	24.5 (4.6)	20.5 (3.6)	0.1
Trail Making Test A	39.4 (11.5)	34.6 (9.8)	0.5
Trail Making Test B	108.4 (61.4)	107.9 (79.4)	1.0
SRT-D	6.3 (2.8)	5.8 (1.5)	0.2
SPART-D	6.8 (1.0)	7.0 (1.4)	1.0
Tower of London	15.8 (5.4)	15.6 (6.6)	0.8
Cognitive Impairment Index	22.5 (3.9)	22.3 (2.4)	0.9

**Table 2 brainsci-11-00637-t002:** Classes of SDMT changes in POMS subgroups who underwent specific and non-specific training.

Classes of SDMT Changes	Specific Training (*n* = 8)	Non-Specific Training (*n* = 8)	*p*-Value
No clinically significant changes	0	4	0.008
Clinically significant worsening	0	2	
Clinically significant improvement	8	2	

## Data Availability

The data presented in this study are available upon reasonable request, only for the purpose of replication of the analyses included in this study and at the discretion of the corresponding author.

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
