# Peer review of "A Randomized Computer-Assisted Rehabilitation Trial of Attention in Pediatric Multiple Sclerosis: A Post Hoc Analysis"

_brainsci, 2021, doi:10.3390/brainsci11050637_

Round 1
Reviewer 1 Report
This is an interesting report of a post-hoc analysis that aimed to assess the robustness of treatment effects, applying the 4-point SDMT cut-off, on the results of a cognitive rehabilitation trial previously completed by this group. Although the report has merit the reviewer has some additional suggestions which may improve the manuscript and increase its impact in the scientific community .
- The SDMT has several parallel forms, were these applied in the rehabilitation study on which the post hoc analysis was based. Practise effects may improve performance on this measure.
- Was this test suitable for 15 year olds or even younger participants as it is used mainly with the adult population? Has this issue been tested? Are there norms for this age group in Italy for this test?
- Were the children of the two groups matched on general intelligence? This variable may impact the outcome on cognition testing. Children with lower intelligence may have lower cognitive performace, poorer cognitive reserve and variability in neuroplasticity mechanisms
- Clinically meaningful changes in children should be measured against real life changes/measues (ecologicaly valid measures). We cannot refer to a "pragmatic" transfer effect in our findings without real life measures
- Please provide a full description of the statistical approach that was utilized
- Are these results useful for everyday clinical practise of clinicians (Peadiatric neuropsychologists, peadiatric neurologists) and why? Are they limited to the Italian population
Author Response
Dear Professor Dr. Stephan D. Meriney,
Thank you for considering our manuscript for possible publication in Brain Sciences.
We are grateful to the reviewers for their careful scrutiny of our paper and their thoughtful comments which have made it possible to improve our manuscript.
We have now revised the manuscript according to their recommendations.
After this, we think that we have addressed the reviewers’ main concerns and have improved the manuscript’s clarity.
As a result, we believe that the manuscript has improved considerably, and we hope that you may re-consider it for publication in Brain Sciences.
Changes in the paper are marked with yellow.
Below, the reviewers’ responses (in italics) are followed by our comments to the reviewers:
Kind regards
Marta Simone, on behalf of the authors.
Here are our comments and responses to the respective questions and criticisms.
Reviewer Comments to Author:
Reviewer: 1
This is an interesting report of a post-hoc analysis that aimed to assess the robustness of treatment effects, applying the 4-point SDMT cut-off, on the results of a cognitive rehabilitation trial previously completed by this group. Although the report has merit the reviewer has some additional suggestions which may improve the manuscript and increase its impact in the scientific community .
The SDMT has several parallel forms, were these applied in the rehabilitation study on which the post hoc analysis was based. Practise effects may improve performance on this measure.
Response: Thank you for this comment, which give us the possibility to better explain the methods of our study. As already stated in the Materials and Methods section of the paper (page 3, lines 94-96), alternative versions of the cognitive tests, at baseline (T0), and within one week following the end of the three months training program (T1) were administered. This also applies to the SDMT. The use of alternative versions of SDMT has mitigated the practice effect, which is always present in some extent when assessing longitudinal changes of cognitive functions.
Was this test suitable for 15-year-olds or even younger participants as it is used mainly with the adult population? Has this issue been tested? Are there norms for this age group in Italy for this test?
Response: We are grateful to the reviewer for these questions. There is an Italian comprehensive longitudinal study reported in three different papers (Amato MP et al. Cognitive and psychosocial features of childhood and juvenile MS. Neurology. 2008 May 13;70(20):1891-7; Amato MP et al. Cognitive and psychosocial features in childhood and juvenile MS: two-year follow-up. Neurology 2010; 75: 1134–1140; Amato MP et al. Neuropsychological features in childhood and juvenile multiple sclerosis: five-year follow-up. Neurology 2014; 83: 1432–1438.) which attests the SDMT use for 15-years-old and even younger patients. It is not properly validated in younger patient but it is the most largely used scale for cognitive evaluation in this population.
Were the children of the two groups matched on general intelligence? This variable may impact the outcome on cognition testing. Children with lower intelligence may have lower cognitive performance, poorer cognitive reserve and variability in neuroplasticity mechanisms
Response: We are grateful to the reviewer for these comments, which give us the possibility to better explain the selection procedure of the patients. Although a general intelligence test has not been administered, patients with a severe impairment on cognitive tasks other than attention or with major psychiatric illness (including intellectual disabilities) were excluded. Moreover, patients were randomized to receive a specific computer training (ST) or to receive a nonspecific computer training (n-ST) with a 1:1 ratio. Randomization was performed by an independent researcher on the basis of a computerized list of random numbers. As a result of the randomization process, at baseline, no differences were found between the 2 treatment arms regarding sex, age, and most importantly in terms of neuropsychological performances.
Clinically meaningful changes in children should be measured against real life changes/measures (ecologically valid measures). We cannot refer to a "pragmatic" transfer effect in our findings without real life measures
Response: We agree with the reviewer for this comment. We have already underlined this limitation in the discussion section; indeed, we did not include in the study protocol the collection of measures of clinically meaningful change such as school performance or parent reported behaviors because this has been a pilot study designed for a fast evaluation of the efficacy of a cognitive training (in treated vs untreated patients) in POMS patients with attention impairment.
Please provide a full description of the statistical approach that was utilized
Response: We greatly thank the reviewer for this suggestion. We have now added, the following, more detailed “statistical analysis” section:
“All analyses were post hoc and not pre-specified.
Continuous variables were described as mean and standard deviation (SD), cate-gorical variables as frequency and percentage. Group comparison has been performed using the Student’s t test, the Mann-Whitney U test and the Fisher's exact test when appropriate.
As previously reported elsewhere [17], by using the mean and SD from the nor-mative values for each test we obtained a global cognitive score, the Cognitive Im-pairment Index (CII). This score allows the evaluation of changes in cognitive perfor-mances independently by the number of cognitive tests failed at the neuropsychological evaluation [17, 19, 21-22]
For each patient, a grading system was applied to individual cognitive tests, based on the number of SDs below the control mean (i.e. grade 0 was given if the patient scored at or above the control mean, 1 if he/she scored below the control mean, but at or above 1 SD below the control mean, and so on until all patient scores were accommodated). [17, 19, 21-22] The individual CII is obtained by the sum of all the patient’s scores. Moreover, to calculate the delta CII, which is the calculated as CII score at T1 – CII score at T0. The higher is the improvement in the global cognitive score as measured by the CII at the end of the training (T1), the lowest is the delta-CII.
To quantify the clinical impact of changes over time of the SDMT in the cohorts of patients exposed to ST and to nST we have applied the following categorization of the delta SDMT scores (delta SDMT=SDMT score at T1 – SDMT score at T0) by using a 4-point SDMT cut-off:
- delta SDMT between -3 and 3 = not clinically significant SDMT change.
- delta SDMT ≤ - 4 = clinically significant SDMT worsening.
- delta SDMT ≥ 4 = clinically significant SDMT improvement.
Thereafter, we compared the proportion of patients reporting a clinically significant SMDT improvement at the end of the 3 months training program by using the chi-square test.
Finally, to evaluate the relationship between changes of the SDMT scores and of the CII, we have compared the delta-CII among the groups of patients stratified on the basis of the delta-SDMT (clinically/not clinically significant SDMT changes).
Statistical analysis was performed by using SPSS software (SPSS, version 22.0; SPSS, Chicago, Ill).”
Are these results useful for everyday clinical practise of clinicians (Peadiatric neuropsychologists, peadiatric neurologists) and why? Are they limited to the Italian population?
Response: Thank you for this comment. These results, if confirmed on larger populations, may be useful for everyday clinical practice to assess the robustness of treatment effects of cognitive rehabilitation trials using SDMT as outcome measure. We have now added some comment about this point in our conclusion.
Reviewer 2 Report
Thank you for this interesting paper on using a cognitive intervention to improve SDMT performance in pediatric MS patients.
1) could you provide a brief description of what was included in the specific computer training exposure.
2) In Lines 115, 116, 118 and 158 – did you mean to say “SDMT”?
3) Could you briefly describe how the scale of the cognitive impaired index is derived and what the numbers in figure 1 mean?
Author Response
Dear Professor Dr. Stephan D. Meriney,
Thank you for considering our manuscript for possible publication in Brain Sciences.
We are grateful to the reviewers for their careful scrutiny of our paper and their thoughtful comments which have made it possible to improve our manuscript.
We have now revised the manuscript according to their recommendations.
After this, we think that we have addressed the reviewers’ main concerns and have improved the manuscript’s clarity.
As a result, we believe that the manuscript has improved considerably, and we hope that you may re-consider it for publication in Brain Sciences.
Changes in the paper are marked with yellow.
Below, the reviewers’ responses (in italics) are followed by our comments to the reviewers:
Kind regards
Marta Simone, on behalf of the authors.
Here are our comments and responses to the respective questions and criticisms
Thank you for this interesting paper on using a cognitive intervention to improve SDMT performance in pediatric MS patients.
1) could you provide a brief description of what was included in the specific computer training exposure.
Response: Thank you for this comment. We have now added in the methods section a brief description of the specific computer training exposure as follow.
“The ST targets focused, sustained, selective, alternating and divided attention and consists of a group of hierarchically organized tasks that exercise different components of attention, proceeding from sustained to selective, alternating and finally divided attention exercises. The sequence of the exercises places increasing demands on complex attention control and working memory systems.”
2) In Lines 115, 116, 118 and 158 – did you mean to say “SDMT”?
Response: We are grateful for this suggestion. We have corrected the mistake in the text.
3) Could you briefly describe how the scale of the cognitive impaired index is derived and what the numbers in figure 1 mean?
Response: We thank the reviewer for this comment which give us the possibility to better explain the methods we have applied and our results. As responded to the reviewer 1 we have now added in the method section a detailed statistical analysis section which includes a detailed description on how we have calculated the cognitive impaired index. Moreover, figure 1 shows that patients who reported a clinically significant improvement of the SDMT scores were those who also reported the major improvement of the global cognitive functions as measured by the CII. On the contrary, patients who did not improve their SDMT performances and those who remained stable on SDMT showed a lower impact on the CII.
Round 2
Reviewer 1 Report
I have no further comments.